# Horner’s Syndrome and Lymphocele Following Thyroid Surgery

**DOI:** 10.3390/jcm12020474

**Published:** 2023-01-06

**Authors:** Mara Carsote, Claudiu-Eduard Nistor, Florina Ligia Popa, Mihaela Stanciu

**Affiliations:** 1Department of Endocrinology, “Carol Davila” University of Medicine and Pharmacy & “C.I. Parhon” National Institute of Endocrinology, 011683 Bucharest, Romania; 2Department 4—Cardio-Thoracic Pathology, Thoracic Surgery II Discipline, “Carol Davila” University of Medicine and Pharmacy & Thoracic Surgery Department, “Dr. Carol Davila” Central Emergency University Military Hospital, 013058 Bucharest, Romania; 3Physical Medicine and Rehabilitation Department, “Lucian Blaga” Faculty of Medicine, University of Sibiu, 550169 Sibiu, Romania; 4Department of Endocrinology, Faculty of Medicine, “Lucian” Blaga University of Sibiu, 550169 Sibiu, Romania

**Keywords:** thyroid surgery, Horner syndrome, thyroid, thyroidectomy, lymphocele, endocrine, medullary thyroid cancer, chylous leakage, eye

## Abstract

Horner’s syndrome (HS), caused by lesions of the 3-neuron oculosympathetic nerve pathway (ONP), includes the triad: blepharoptosis, miosis and anhidrosis (ipsilateral with ONP damage). Thyroid–related HS represents an unusual entity underling thyroid nodules/goiter/cancer–HS (T-HS), and post-thyroidectomy HS (Tx-HS). We aim to overview Tx-HS. This is a narrative review. We revised PubMed published, full-length, English papers from inception to November 2022. Additionally, we introduced data on post-thyroidectomy lymphocele/chylous leakage (Tx-L), and introduced a new pediatric case with both Tx-HS and Tx-L. Tx-HS: the level of statistical evidence varies from isolated case reports, studies analyzing the large panel of post-thyroidectomy complications reporting HS among the rarest side effects (as opposite to hypocalcemia), or different series of patients with HS due to various disorders, including T-HS/Tx-HS. Tx-HS is related to benign or malignant thyroid conditions, regardless the type of surgery. A pre-operatory rate of T-HS of 0.14%; a post-operatory rate of Tx-HS between 0.03% and 5% (mostly, 0.2%) are identified; a possible higher risk on endoscopic rather than open procedure is described. Incomplete HS forms, and pediatric onset are identified, too; the earliest identification is after 2 h since intervention. A progressive remission is expected in most cases within the first 2–6 months to one year. The management is mostly conservative; some used glucocorticoids and neurotrophic agents. One major pitfall is an additional contributor factor like a local compression due to post-operatory collections (hematoma, cysts, fistula, Tx-L) and their correction improves the outcome. The prognostic probably depends on the severity of cervical sympathetic chain (CSC) lesions: indirect, mild injury due to local compressive masses, intra-operatory damage of CSC like ischemia and stretching of CSC by the retractor associate HS recovery, while CSC section is irreversible. Other iatrogenic contributors to HS are: intra-operatory manipulation of parathyroid glands, thyroid microwave/radiofrequency ablation, and high-intensity focused ultrasound, and percutaneous ethanol injection into thyroid nodules. Tx-L, rarely reported (mostly <0.5%, except for a ratio of 8.3% in one study), correlates with extended surgery, especially lateral/central neck dissection, and the presence of congenitally—aberrant lymphatic duct; it is, also, described after endoscopic procedures and chest-breast approach; it starts within days after surgery. Typically low-fat diet (even fasting and parental nutrition) and tube drainage are useful (as part of conservative management); some used octreotide, local sealing solutions like hypertonic glucose, Viscum album extract, n-Butyl-2-cyanoacrylate. Re-intervention is required in severe cases due to the risk of lymphorrhoea and chylothorax. Early identification of Tx-HS and Tx-L improves the outcome. Some iatrogenic complications are inevitable and a multifactorial model of prediction is still required, also taking into consideration standardized operatory procedures, skillful intra-operatory manipulation, and close post-operatory follow-up of the patients, especially during modern era when thyroid surgery registered a massive progress allowing an early discharge of the patients.

## 1. Introduction

Horner’s syndrome (HS), caused by the lesions of the 3-neuron oculosympathetic nerve pathway (ONP), includes the classical triad of eyelid ptosis (blepharoptosis), miosis (with consecutive anisocoria) and anhidrosis on the same side of the face, ipsilateral with ONP lesion [1,2,3,4].

Clinical features and circumstances of onset are connected to ONP anatomy that starts from hypothalamus and synapses at cervicothoracic (C8-T2) spinal cord at the level of intermediolateral grey matter [1,2,3,4]. Other elements are: apparent enophthalmus, facial flushing (due to vascular dilation), headache, heterochromia of the iris (in some congenital forms of HS), etc. [1,2,3,4]. Acute painful onset (even with sudden visual loss) embraces an emergency scenario [5]. Tadpole pupil (segmental spasm of the iris dilator muscle) is identified as spontaneous condition, but in 46% of cases it accompanies HS [6]. Historically, Swiss ophthalmologist Johann Friedrich Horner described the syndrome in 1869, as, well as French physiologist Claude Bernard (in 1952) and others, that is why the complete name is Claude-Bernard-Horner syndrome, yet, “Horner’s syndrome” remains the most commonly used term now days [7,8].

Sudden onset of HS on an apparently healthy person without a clear index of suspicion requires a multidisciplinary evaluation including an ophthalmological, otorhinolaryngological, and neurological evaluation, and, also, imaging of the entire ONP in order to check the brain, orbits, neck, and chest [9,10]. A multitude of causes (acquired or congenital) are reported, from traumatic, tumour, inflammatory disorders to stroke as well as routine activities like “hairdresser syndrome” causing vertebro-basilar insufficiency, etc.; some of them are iatrogenic, mostly after surgery in both children and adults; remarkably, an important subgroup has a certain potential of full recovery [11,12,13,14]. 

Traumatic HS is mainly caused by cervical spine injury or carotid artery dissection [15,16]. Unusual traumatic entities have been reported. For instance, we mention a case from 2021 concerning a middle-third clavicle fracture which is the most frequent type of fracture of this bone that, yet, exceptionally complicates with HS [17]. Also, tumours (like osteocondroma), traumas or surgical interventions at the level of first rib might cause the condition [18,19]. Cranio-cervical artery dissection, usually spontaneously arising or after minor trauma causes stroke in young adults, but, also, HS in 25% of cases [20,21].

Infectious and inflammatory circumstances might cause HS [22,23,24,25]. We mention: giant cell arteritis, a systemic vasculitis affecting arteries with medium and large diameter (despite the underling mechanisms not being clearly understood) [22]; unusual cases like central nervous system infection with Toxoplasma in a HIV positive subject [23]; retropharyngeal abscess [24]. Another unusual condition is spinal epidural abscess that typically is diagnosed based on the triad of back pain in association with fever and neurological deficiencies, HS not being a classical presentation [25]. Lesions or removal of cervical sympathetic chain (CSC) for different causes vary from frequent ones like Pancoast tumour to less frequent as schwannoma originating from CSC or cervical spontaneous intradural disc herniation [26,27,28,29]. 

Iatrogenic panel is heterogeneous, including: peripheral nerve blocks; anterior supraclavicular or interscalene approach [30,31,32,33,34]; internal jugular vein catheterization [35]; cardiac sympathetic denervation for some severe arrhythmias [36]; thoracoscopic resection of thoracic inlet neuroblastic tumours might complicate with HS [37,38]; thoracotomy for paediatric neuroblastoma has even a higher incidence of HS [39,40]. A systematic review on complications associated with anterior cervical spine surgery identified through 240 articles a rate of 0.4% concerning HS belonging to a panel of side effects that varied from frequent ones (like adjacent segment disease of 8.1%, dysphagia—5.3%, cervical C5 palsy—3%) to the rarest as oesophageal perforation (0.2%), vertebral artery injury (0.4%), etc. [41]. Another study on anterior cervical discectomy and fusion (representing the most frequent surgical procedure at the level of cervical spine in US) showed a prevalence of HS between 0.06% and 1.1% [42,43]. Another modern iatrogenic circumstance is post—lung transplant HS [44].

Post-operatory HS is often under-diagnosed; it associates a great potential of reversibility, while the actual recognition may frequently be established after the patient was discharged for a specific surgery like thyroidectomy [30,31,32,33,34,35,36,37,38,39,40,41,42,43,44]. 

### Aim

Our purpose is to overview HS in relationship with thyroid status, particularly, after thyroidectomy and, also, with regard with post-operatory development of a lymphocele.

## 2. Methods

This is a narrative review. We revised PubMed published full-length, English papers concerning the mentioned topic, from inception to November 2022. The research words are “Horner syndrome” (alternatively, lymphocele or chylous leakage) and either “thyroid” or “thyroidectomy”. Also, we introduce a prior unpublished case of pediatric post-thyroidectomy HS in association to post-operatory development of a lymphocele.

## 3. Thyroid Conditions and HS and Lymphocele/Chylous Leakage

### 3.1. Thyroid Disorders and HS: From Goiter to Iatrogen Elements

#### 3.1.1. Thyroid Aspects Causing HS

Thyroid—related HS represents a rare cause of HS among large panel of conditions that we previously mentioned, including thyroid nodules/goiter/cancer—associated HS, respective post-thyroidectomy HS. A certain index of suspicion is necessary, thus the importance of awareness despite limited epidemiological impact. The first two issues concerning thyroid pathology/surgery and a newly diagnosed case of HS are: whether HS is strictly related to this endocrine and endocrine surgery field or, in fact, there is another synchronous disorder to induce HS; and, secondary, once that thyroid involvement is confirmed, is there any place of intervention in order to improve the outcome. As initial step of assessment, pharmacological tests with cocaine, and aproclonidine in adults might help the diagnostic, but usually imaging procedures are helpful on daily practice in addition to clinical, especially ophthalmological, but, also, neurological evaluation [45,46,47].

HS as first presentation of a large thyroid lump/goiter/cancer is due to CSC compression; the level of statistical evidence remains a few case reports [48,49,50]. Both benign and malign nodular thyroid conditions may present HS (one case of Riedel’s thyroiditis is reported) [51,52,53,54]. Remarkably, one adult female had unusual brain metastasis (in addition to cervical and lung spreading) from a papillary thyroid carcinoma with HS as onset element since HS—related ONP lesions might be presented not only at neck level [55].

#### 3.1.2. Thyroid Surgery Followed by HS

Of historical note, a literature review from 2002 evidenced 38 cases and added another one concerning HS—associated thyroid neoplasia (8 out of 39 were malignant diseases) [56]. On PubMed, the first paper addressing CSC damage during surgeries for thyroid disorders was published in 1965 [57]. The first ever case of post-thyroidectomy HS dates from 1915 [58].

The statistical evidence on post-thyroidectomy HS includes isolated case reports, various studies that analyzed a large panel of post-thyroidectomy complications and reported HS among these side effects; other type of papers includes different series of patients with HS that followed the underling disorders and identified thyroid or thyroid surgery issues among the bigger picture of non-thyroid causes. 

All types of thyroidectomy are reported to potentially add a risk of developing post-operative HS, from open to endoscopic procedures, including particular resections of lymph nodes and reverse “L” thoracotomy for tumors situated at the level of cervicothoracic junction, including aggressive thyroid carcinoma. Thyroid surgery for either benign or malign conditions (like differentiated and anaplastic cancer, as well as medullary thyroid carcinoma) might induce HS [59,60,61,62,63,64].

Post-surgery identification of partial and complete HS is reported within hours to days, with the highest ratio in one study of 5%, but most studies agree on less than 1%. The mechanisms of post-thyroidectomy HS are local post-operatory liquid collections (neck ultrasound representing the first line tool of investigation under such circumstances), intra-operatory damage of SCS, including ischemia-induced lesions, and stretching of SCS by the retractor, and, probably, particular anatomic configuration of SCS [59,60,61]. Also, we mention thyroidectomy-associated exploration of parathyroid glands in addition to a SCS ganglioneuroma/paraganglioma/schwannoma removal [65,66,67]. One retrospective study on 9 cases of HS out of 2208 surgeries for thyroid and parathyroid conditions identified a 0.14% pre-operative rate (N = 3 cases; N1 = 2 with benign conditions, N2 = 1 case of anaplastic cancer), respective a 0.27% post-surgery rate of HS; patients’ ages varying between 22 and 87 years [68].

Tang M. et al. published in 2022 the results of 1213 thyroidectomies pointing out that HS ratio after endoscopic approach is higher when compare to open procedure (0.39% *versus* 0.29%) [69]. Endoscopic thyroid surgery (ETS)—related HS is reported in a few cases reports. For instance, we mention a 31-year-old woman who had a total thyroidectomy done in addition to central lymph node dissection via ETS. Pathological report confirmed a papillary thyroid microcarcinoma with an autoimmune chronic background. She developed HS within the third post-operatory day with progressive remission after a 3-month period of surveillance [70]. Another 34-year-old woman with papillary thyroid carcinoma and Graves’s disease had a left-side minimally invasive procedure of video-assisted thyroidectomy and neck dissection complicated with HS second day after the intervention. The patient was treated with glucocorticoids and neurotrophic drugs, and HS remitted within one year [71]. Another series on 16 patients with papillary thyroid cancer who underwent bidirectional approach of video-assisted neck surgery identified one case of HS (1/16) [72].

Parapharyngeal space manipulation is particularly prone to HS [73]. Thus, another surgical procedure that has been identified as causing HS is the resection of parapharyngeal lymph nodes metastasis from thyroid cancer (and even one case of parapharyngeal mass underling an ectopic thyroid with post-operatory HS is reported) [74]. This is the subject of a study on 97 patients who had this type of malignancy spreading. The profile of post-operatory side effects showed dysphagia in 5% of cases, glossal deviation in 3%, and HS in 2% of the individuals [75].

Early identification of HS after thyroid surgery might increase the potential of reversibility [76,77]. Of note, we mention the case of a 27-year-old male who suffered a total thyroidectomy with lateral neck dissection for a papillary thyroid carcinoma and developed HS within the first 2 h after surgery (partial remission after 2 months under conservative management) [78]. This is the earliest identification of HS after thyroid surgery. Most cases of post-thyroidectomy HS were treated conservatively with progressive (complete or partial) remission within months (up to one year) [79,80]. As early intervention, removal of local hematoma and post-surgery liquid collections is helpful. Corticotherapy was offered to some patients [71,81]. For example, this is a 27-year-old female who developed HS second day after thyroidectomy for a benign thyroid condition. Except for a 2-week regime of prednisolone, no intervention was added, with improvement within 6 months [81]. Neurotroph factors, respective mecobalamine represented an alternative [71,82]. For example, this is a 44-year-old woman developing HS, particularly, miosis and ptosis, after thyroid ablation; mecobalamin was used immediately after surgery and an incomplete remission was registered within following 5 months [82]. Generally, direct injury of SCS or removal of hematomas or local cysts might expect full recovery, while complete SCS section most probably will induce a persistent HS. A correlation of recovery potential with underling malignant/benign thyroid condition or open/video-assisted/endoscopic procedure is difficult to be established, but we consider that probably there is a multifactorial model [83,84].

Iatrogenic HS with regard to thyroid field additionaly involves thyroid microwave ablation [82,85,86,87]. A systematic review on radiofrequency ablation for benign thyroid nodules including 32 studies (N = 3409 patients) showed the safety of the procedure; HS was identified immediately after procedure in one case (the syndrome improved after 6 months without any therapy, but persisted more than 6 months) [87]. A higher risk concerns thyroid nodules that are located close to the middle cervical sympathetic ganglion (situated near inferior lobe of the thyroid and lateral to common carotid artery), and ablation—induced hematoma or thermal damage [87,88]. One systematic review and meta-analysis (including 9, respective 6 studies) from 2022 revisited the studies between 1990 and 2021 concerning high-intensity focused ultrasound as a therapeutic approach for benign thyroid nodules. In addition to common side reactions like local pain and tegument changes, major side effects included 0.014% of patients with transient vocal cords paralysis and dysphonia, but, also, 0.5% of subjects experiencing transitory HS [89]. HS after this type of procedure might be symptomatic (for instance, neck pain that requires other investigations for differential diagnostic) [90]. A multicenter, 3-year European study followed 65 patients after the same approach (mean age of 51.1 years, 86.2% females) and identified one case of HS (1.5%) [91]. One study on 90 patients to whom percutaneous ultrasound—guided microwave ablation was performed identified another case of HS (1.3%) [92]. Another study on 121 subjects who underwent a similar intervention for benign nodules reported one patient with HS (1%) that remitted after 2 months [93]. Another study on medically refractory Graves’s disease (N = 30) pointed out a HS rate of 6.7% [94]. Another iatrogenic HS is due to percutaneous ethanol injection into thyroid nodules [95].

### 3.2. Thyroidectomy/Lymph Nodes Dissection and Lymphocele/Chylous Leakage

Generally, lymphocele (or lymphocyst), a cyst filled with lymph fluid, without an inflammatory or granulomatous reaction, occurs after different surgical procedures within first weeks [96,97,98,99,100,101,102,103,104,105]. Post-operatory lymphatic leakage may be of lymphorrhea type (for instance, a lymphocele) or of chylorrhea type (with different forms of chylous leakage) [96,106].

Lymphocele is a self–limiting condition in majority of cases; 5–18% of individuals associate local pain, infection, persistent lymphorrhea, local compression, thus surgical intervention is required in these symptomatic cases and those with no response to conservative approach, starting with drainage preferably through imaging guiding (there is no general consensus on optimal therapy which mostly depends on the site and prior surgical and medical co-morbidities) [96,97,98,99,100,101,102,103,104,105,106]. A higher risk of lymphocele after different surgeries relates to advanced age, increased body mass index, numerous resected lymph nodes, open surgery rather than laparoscopic approach, and procedures at the level of anatomic areas with a high density of lymphatic channels (thyroid surgery not being one of the mostly known locations with such a risk) [96,97,98,99,100,101,102,103,104,105,106].

Post-thyroidectomy lymphocele/chylous leakage are exceptional, being correlated with thyroid conditions with malignant potential, extended surgery, congenitally—aberrant lymphatic duct; left side is mostly affected due to damage of left-sided thoracic duct [103,104,105,107]. However, lymphocele following right thyroidectomy is also reported in relationship with large thyroid masses or right neck dissection [103]. Also, neck lymph node resection, both central and lateral, seems a more important contributor that thyroid removal itself [103,108,109,110]. 

Lymphatic/chylous leakage develops within days after thyroidectomy; the approach depends on severity of the clinical manifestations, lymphocyst size, and compressive elements, varying from low fat diet (including complete restriction of enteral feeding or fasting with exclusive parenteral feeding), use of somatostatin analogues for reducing chylous production to negative pressure drainage, and even re-intervention [103,104,105,108,109]. The drainage of post-operatory cervical lymphocele remains the major step of practical approach [104]. One study reported an incidence of lymphatic leakage of 0.5% following thyroid and/or lymph nodes resection for benign and malign conditions. 66% of patients responded to fasting, while the others required re-intervention [111]. Percutaneous imaging (mostly, ultrasound)—guided techniques helps the identification of the leakage site [112]. Ethanol sclerotherapy is rather on historical note [105]. A series of 6 cases introduced thoracic duct embolization due to leakage after neck dissection [113]. However, after this procedure, there is a risk of developing foreign body granulomatous lymphadenitis [114].

A serious injury of an occult congenitally-aberrant lymphatic duct is mostly a candidate to re-intervention [103]. Dramatic damage of thoracic duct might complicate with lymphorrhoea, and chylothorax, thus the importance of adequate decision of re-surgery [108]. We mention such a case where the patient refused re-intervention and was treated with an novel approach. In 2021, the first report of local injection with hypertonic glucose for a refractory chylous leakage on a 55-year-old female who suffered a total thyroidectomy with central and bilateral neck lymph node dissection for thyroid cancer. The neck swelling spontaneous developed after two days since surgery; the liquid was evacuated through ultrasound guided aspiration, the patient was not responsive to low fat diet and compressive bandaging with daily ultrasound-guided aspiration for more than 30 days, she received a drainage tube followed by intravenous somatostatin to decrease the production of lymphatic fluid, and, finally (since the patient refused re-intervention) local injection of hypertonic glucose via drainage tube sealed the site with a good outcome [109]. Another innovative approach was the first time use of Viscum album extract for damage of thoracic duct after modified radical neck dissection on a 54-year-old female with papillary thyroid cancer and bilateral lymph nodes metastasis. Initially, surgical procedures in addition to ocreotide and parenteral feeding were performed, however, unsuccessful [115]. An alternative is the use of n-Butyl-2-cyanoacrylate to seal the tissues with respect to lateral neck dissection [116].

We mention a few studies reporting chylous/lymphatic leakage, most of them with a small size considering the enroled population. A series of 12 subjects concerning lateral neck (Vb) dissection via endoscopic surgery with chest-breast approach (CBA) identified one subject (1/12) with the complication [117]. A study on 57 patients who underwent left central lymph node dissection via thyroidectomy (CBA) for thyroid cancer found one case (belonging to the subgroup with endoscopic, not open approach) [118]. Another study (N = 24) on patients referred for lateral neck dissection at the levels IIA, IIB, III, and IV via endoscopic CBA for malignant thyroid conditions identified 2 cases with chylous leakage (8.3%) [119]. Another retrospective study on 18 subjects with endoscopic thyroidectomy and neck dissection (II, III, IV, and VI) versus 20 patients with open total thyroidectomy showed a similar prevalence of lymphatic drainage (1/18 versus 3/20, *p* = 0.606) [120]. A study reported an incidence of 0.9% with respect of chylous drainage; 43.7% of patients had central neck lymph nodes dissection (without lateral neck dissection) and they were mostly conservatively treated; the authors suggest surgical management only if the liquid does not decrease more than 50% after two days of diet and drainage [121]. Another series of two patients developed chylous leakage after central lymph nodes dissection for a thyroid malignancy; they were treated with a conservative method combing compression bandage with negative pressure drainage and external fixation of neck brace in addition to with low-fat diet and experienced an improvement after 24–48 h and a complete remission within 10 days [122]. A retrospective study on 13,224 thyroidectomies identified 20 chylous fistulas (7/20 were already diagnosed with lymphatic leak at the moment of admission). Nutritional intervention and drainage were helpful, but some required re-intervention (duct ligation) [123].

Lymphocele as a distinctive post-thyroidectomy complication is referred on a minority of case reports. We extended the research on the larger category concerning post-thyroidectomy chylous leakage which, generally, is recognized as a complication with a low incidence that requires a mix management (diet + drainage + selective re-intervention). These complications belong to the complex picture of thyroid surgery, regardless modern approach as CBA or programed neck dissection, and they are less related to the underling endocrine disorder on terms of hormonal and autoimmune assays rather to the proliferative potential and lymph node spreading. A model of prediction concerning such complications is far from being implemented in daily practice while post-operatory surveillance and decision making is multidisciplinary, and, as opposite to other surgeries, is rather based on individual decision through a day by day surveillance. 

### 3.3. Post-Thyroidectomy HS and Lymphocele

In addition to literature review, we introduce a pediatric case that has not been *in extenso* published before contributing to a better understanding of the limited data we have so far concerning this particular topic.

This is a 15-year-old female harboring the RET gene mutation (exon 11, pCys634Gly), consistent with familial Multiple Endocrine Neoplasia (MEN) type 2A syndrome. She was admitted 2 years ago after a total thyroidectomy with central and lateral lymph nodes dissection for a medullary thyroid cancer. Histological report confirmed a bilateral medullary microcarcinoma of 4.5/5 mm on the left lobe and of 3/3 mm on the right lobe with C-cell hyperplasia (pT1N0MO); while immunohistochemistry showed a Ki67 proliferation index of 1–2%, and positive CROMO. Post-operatory hormonal assays confirmed the normalization of pre-operatory 5-fold increase of serum calcitonin, with no other endocrine elements of MEN2A; iatrogenic post-surgery hypothyroidism was controlled under adequate daily levothyroxine substitution; she experienced transitory hypocalcemia. Second day after rapid discharge (one-day hospitalization for thyroidectomy), she first described mild droppy right eyelid followed for the next days by right eye myosis, persistent dry eye sensation, conjunctival discomfort, and episodes of alternative nasal congestion and obstruction which were actually the most disturbing clinical elements. (Figure 1).

The diagnostic of HS was established after she was seen by different specialists as outpatient; she started local therapy with artificial tears and local nasal products with no significant improvement. In the meantime, a right neck swelling was progressively increasing, thus she was re-admitted 5 weeks since thyroidectomy. Neck ultrasound revealed a hypoechoic cystic-like area at the level of right lateral cervical level (of 60 mm maximum diameter) without thyroid remnants or local lymph node enlargement. (Figure 2).

Neck computed tomography confirmed bilateral, lateral cervical, well-shaped masses with cystic appearance, of oval form (of 40/25/57.4 mm on the right, respective of 1.6/27/40 mm on the left). (Figure 3). 

Ultrasound—guided fine need aspiration of the right mass provided cvasi-complete reduction of the liquid with stationary ultrasound features after 90 min. The liquid examination showed mature lymphocytes, rare erythrocytes, rare cholesterol crystals, lymph fluid, consistent with the diagnostic of lymphocele. A mild clinical improvement of HS—associated features was registered, but within 2 days, the neck mass rapidly regrew, and the patient was re-admitted. Under local anesthesia, there was a liquid evacuation (macroscopic aspect with milky-like aspect, a volume of 5–6 mL) and a drain tube was placed for 4 days. Ultrasound aspects showed a progressive regression of the right cystic mass with improvement of HS which slowly remitted within the next 2–3 months. (Figure 4). 

Despite full recovery, intermittent episodes of nasal obstruction/congestion (which were never registered pre-operatory) persisted for one year. The patient remained under life time protocol of surveillance for MEN2.

## 4. Discussion

HS concerns a dramatic number of medical and surgical practitioners with respect to various causes located at head, neck, and thorax that are presented from birth to elderly seniors; the level of statistical evidence depending on the underling disorder, but, numerous entities are limited to several case reports [124,125,126,127,128,129,130,131,132,133,134].

### 4.1. Integrating HS and Lymphocele to the Panel of Post-Thyroidectomy Complications

When it comes to thyroid picture—associated HS and lymphocele, especially post-thyroidectomy HS, a few data are published so far. Of note, we prior mentioned the studies where these complications were identified, but, there are many other cohorts without any such complications [135]. Under these specific circumstances, HS and even chylous leakage play only a small role on the large ensemble regarding the complications following thyroid surgery [136,137,138].

The most frequent complications, a part from iatrogenic hypothyroidism, are hypocalcemia followed by lesions of recurrent laryngeal nerve/vocal folds. We mention three studies that reported HS (and even chylous/lymphatic fistula/leakage in two of them). One retrospective cohort on 456 thyroidectomies (a 10-year experience) showed: asymptomatic, respective clinically symptomatic hypocalcemia (23.9%, respective 10.9%), vocal fold palsy (3.7%) as the most frequent, while hematoma was identified in 0.65% of cases, seroma in 1.3% and HS in one case (representing 0.2%) [136]. A study on 3000 patients who underwent robotic thyroidectomy with axillary approach for thyroid cancer identified the most common complication hypocalcemia (37% of persons had transitory type and 1% of cases were permanent); local collections like seroma (1.73%), hematoma (0.37%), chylous leakage (0.37%), and HS in 0.03% of all cases [137]. Another study on 2636 individuals with thyroid cancer showed post-operatory complications associated with the extent of the procedure, as following: hypoparathyroidism (28.7 %) as the most common side effect, as well as, vocal cord palsy (0.9%), seroma (4.7%), chylous fistula (1.8%), hematoma (0.5%), and, also, HS (0.2%) [138].

A skillful approach is mandatory in any thyroid surgery; however, some post-operatory events are inevitable, even nowadays with an impressive progress registered within this field, thus the level of knowledge remains essential in order to recognize and provide the adequate management, to avoid cosmetic side effects of HS—associated eye anomalies, the metabolic, fluids and local complications of large post-operatory collections/fistula, to overall improve the quality of life after a surgical procedure that already is performed with a minimum days of hospitalization [139,140,141].

Any of the two mentioned complications increases the duration of hospitalization or readmission/re-evaluation rates to different departments, thus their importance.

### 4.2. Pediatric HS

Thyroid—related HS, either surgical or not, is exceptional in children and teenagers. We reported a pediatric case of post-thyroidectomy HS with regard with medullary thyroid cancer with a secondary contributor to HS as lymphocele (in addition to surgical act itself) that required meticulous post-operatory surveillance and a certain level of awareness after procedure that, otherwise, was uneventful. Moreover, the surgery and following readmissions were done within the first year of COVID-19 pandemic where new protocols restricted the access to hospital and a fast decision was needed in order to provide the best outcome and avoid unnecessary prolonged hospitalization under these delicate circumstances. All over the world, a re-shaped window of access to investigations and surgery, including endocrine surgery, was registered especially during the first pandemic waves; non-emergency interventions being difficult to be stratified [142,143,144].

We identified a similar post-operatory pediatric case of HS: a 15-year-old girl who underwent a total thyroidectomy for a benign condition (with unidentified pre-operatory proliferative potential according to the fine needle aspiration); she further developed right HS during the second day after the procedure. HS recovered within 6 months [145]. Another series of 16 children with papillary thyroid carcinoma and associated surgery identified one HS (1/16) as post-operatory complication [146]. The first pediatric case with papillary thyroid carcinoma and HS as onset was a 14-year-old child (a case report from 2010) [147]. One series on 26 children with differentiated thyroid malignancy were followed for a median of 14.2 years and 2/26 cases developed post-operatory HS [148].

Also, Mayo Clinic experience of pediatric thyroidectomy (N = 177) reported one transitory HS (0.6%) [149]. Of note, we identified two publications specifically addressing HS and medullary thyroid carcinoma. One recent case of post-thyroidectomy—related HS and medullary cancer was a 73-year-old female (HS onset from first post-operatory day and recovery was within one month) [150]. The oldest study on this type of cancer and HS dates from 1990; 11 patients with this specific neoplasia were followed for 15 months and identified 3/11 HSs [151]. (Table 1).

Post-thyroidectomy HS belongs to the larger group of pathological entities inducing HS in young patients. Generally, the most common pediatric causes of HS are neuroblastoma, and birth trauma; yet, one third of individuals will remain without an obvious cause (but meticulous investigations are required anyway) [152,153,154,155]. The pediatric approach depends on circumstances from congenital syndromes with severe neurological damage to spontaneous mild detection of an eyelid ptosis or anisocoria; for infant anisocoria the gold standard of investigations remains cocaine testing which is needed in association with neuroimaging evaluation [156,157,158,159]. One systematic review identified 8 studies on pediatric HS (N = 152 patients, aged between birth to 20-year-old) with newly onset forms without a prior medical history; 17/152 were caused by space-occupying lesions (12/17 were neuroblastomas) [160]. A nationwide, population—based Korean study on pediatric (N = 139, 59.7% males) and adult (N = 1331, 51% males) cases with HS showed a cumulative incidence of 2.12 per 100,000 persons, respective of 2.95. For children, the peak ratio was between birth and 4 years; the most frequent tumor was neuroblastoma, while for adults where thyroid tumors [161].

Pediatric HS often displays an atypical presentation like incomplete picture or intermittent presence of the clinical elements, requiring a certain level of awareness [162]. One study on 61 pediatric cases of anisocoria or HS showed that out of 10 cases with HS, 40% were atypical. The studied identified an overall incidence of pediatric HS of 2.54 per 100,000 persons (Northern Ireland) [163]. Partial forms of HS are reported in adults, as well. For instance, the most frequent caused of ptosis, identified through an assessment of ocular motility and pupillary examination, are third nerve palsy and HS; at the other end of the pathogenic spectrum, among the rarest causes, miastenia gravis might onset with ptosis as isolated sign [164,165,166,167]. Blepharoptosis in children may be acute or chronic, progressive or not, isolated or syndromic, congenital or acquired [168].

## 5. Conclusions

To our knowledge, this is first case report of a teenager with HS after surgery for medullary thyroid carcinoma, and the first ever case with both HS and lymphocele after thyroid surgery, and, also, we did not identify any another pediatric case with lymphocele after thyroidectomy and neck lymph nodes dissection; neither found any specific information concerning both HS and lymphocele/chylous leakage in MEN2A patients.

As general note, early identification of HS and potential reversible factors might improve the outcome, while lymphocele/chylous leakage represents a very rare and unusual complication after thyroidectomy and lymph nodes resection. Both topics involve a rather limited amount of published data so far. Some iatrogenic complications are inevitable and a multifactorial model of prediction is still required, also taking into consideration standardized operatory procedures, skilful intra-operatory manipulation, and close post-operatory follow-up of the patients, especially during modern era when thyroid surgery registered a massive progress with very early discharge of the patients.

## Figures and Tables

**Figure 1 jcm-12-00474-f001:**
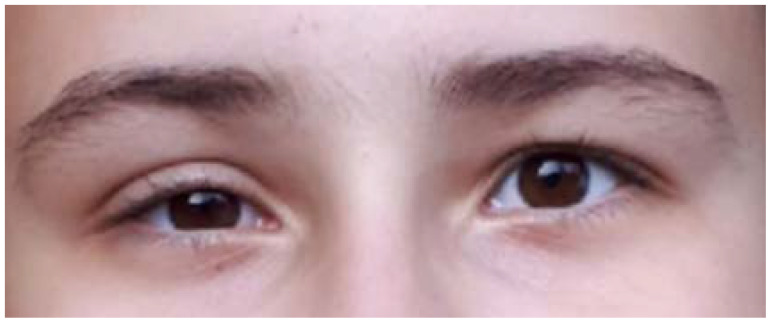
Right Horner syndrome after total thyroidectomy and lateral and central neck for medullary thyroid carcinoma.

**Figure 2 jcm-12-00474-f002:**
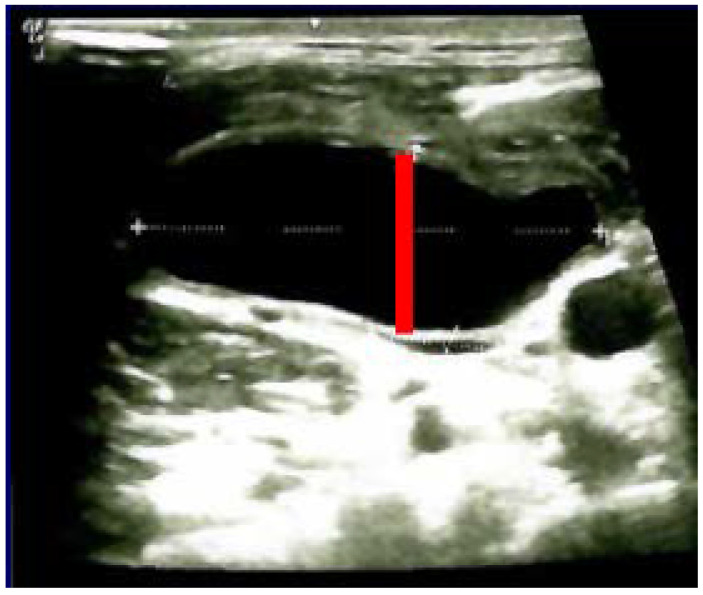
Neck ultrasound (right lateral area): hypoechoic, cystic- like, well-shaped mass of 6 by 2 by 4.5 cm (post-thyroidectomy capture: after 5 weeks since surgery before any additional intervention; red line represents the limits of the mass).

**Figure 3 jcm-12-00474-f003:**
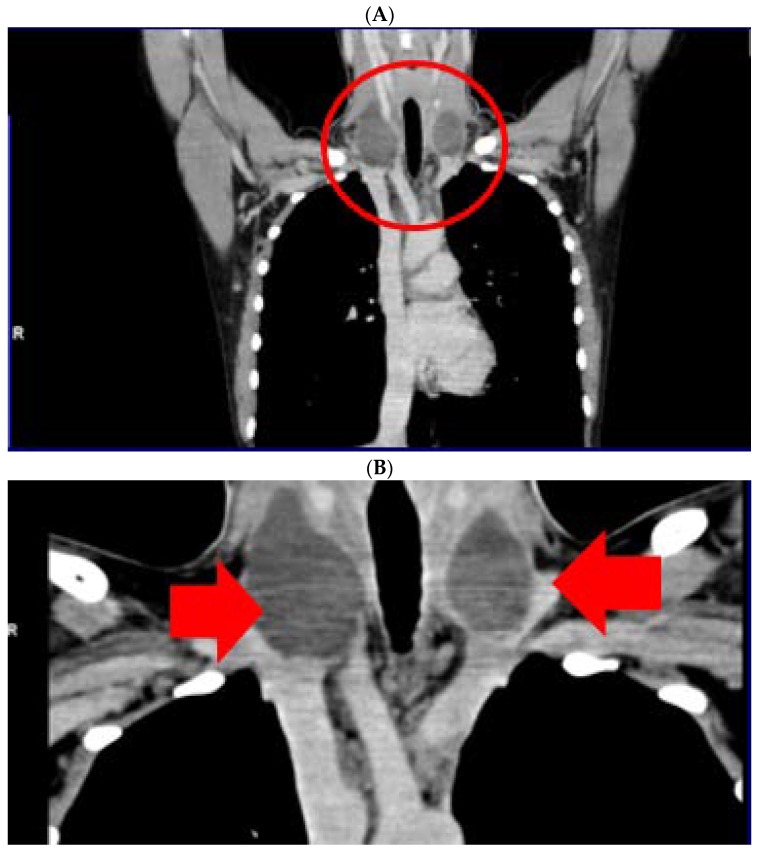
Computer tomography of neck area showing bilateral lateral cervical well-shaped masses with cystic features of 4 by 2.5 by 5.74 cm on the right, respective of 1.6/2.7/4 cm on the left (5 weeks since surgery, before any local intervention). (**A**). Frontal plane—first section; (**B**). Frontal plane—second section; (**C**). Transverse plane; (**D**). Sagittal plane.

**Figure 4 jcm-12-00474-f004:**
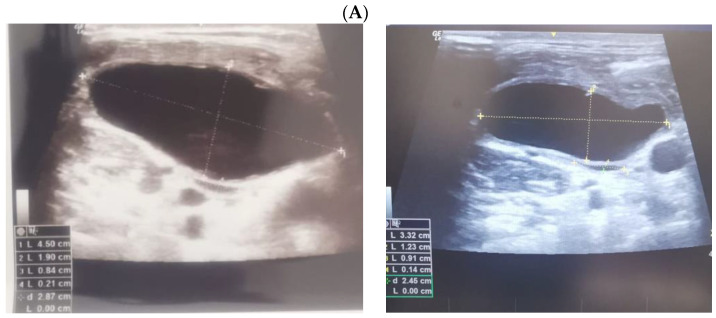
The evolution of within following weeks after thyroid surgery ultrasound features. (**A**). Right neck ultrasound aspect at the moment of fine needle aspiration (5 weeks since surgery)—on the right and 6 days after fine needle aspiration procedure (after cvasi-complete evacuation, the cystic mass relapsed within 48 h)—on the left. (**B**). Right neck ultrasound within the first day of tube drainage (6 weeks since initial surgery) showing a massive reduction of the cystic mass. (**C**). Drainage tube was removed after 4 days. Post-thyroidectomy scar and post-tube removal aspect. (**D**). Right neck ultrasound after 2 weeks from removing the drainage—persistent cystic mass, but with progressive improvement of HS (approximately 9 weeks since initial surgery). (**E**). Right neck ultrasound after another 2 weeks from removing the drainage –progressive reduction of cystic mass, but with remarkable improvement of HS (approximately 11 weeks since initial surgery). (**F**). Scar aspect 12 weeks since surgery and 5 weeks since lymphocele drainage. (**G**). Recovery of HS—approximately 8 weeks since drainage. (**H1**,**H2**). Bilateral neck ultrasound 20 weeks since thyroid surgery showing regression of the lesions on both sides versus prior examinations. (**H1**). Right later-cervical cystic collection of 2 by 2.8 by 0.9 cm (2.5 mL) and peripheral areas of fibrotic reorganization. (**H2**). Left later-cervical hypoechoic, inhomogeneous mass (no vascular signal) showing fibrotic reorganization (0.25 mL).

**Table 1 jcm-12-00474-t001:** Published papers concerning post-thyroidectomy Horner syndrome (the order of display is according to the main text body; please see references number: [59,60,61,62,63,64,65,70,71,72,74,75,76,77,78,79,80,81,82,83,84,85,145,146,148,149,150,151].

Reference Number within Main Text	Studied Population	Thyroidectomy	Central Neck Dissection
**Adult population**			
[59]	case report (1 patient with HS)	+	
[60]	cases series (6 patients with HS)	+	
[61]	case series 21 patients with different tumors (1 patient with HS)	reverse “L” surgical approach	
[62]	case report (1 patient with HS)	+	Selective neck region VI dissection
[63]	retrospective study 1000 patients with thyroid cancer (1 patient with HS)	robot-assisted endoscopic thyroid surgery using a gasless, transaxillary approach	Ipsilateral central compartment node dissection (malignant cases)
[64]	retrospective study 338 patients (1 patient with HS)	robot-assisted endoscopic thyroid operations using a gasless, transaxillary approach	Ipsilateral central compartment node dissection (malignant cases)
[65]	case report (1 patient with HS)	left hemi-thyroidectomy in addition to ectopic parathyroid gland removal and four-gland parathyroid exploration (no pre-operatory imaging parathyroid adenoma localization)	
[70]	case report (1 patient with HS)	+(endoscopic thyroid surgery)	central lymph node dissection (endoscopic thyroid surgery)
[71]	case report (1 patient with HS)	+left-side minimally invasive video-assisted thyroidectomy	neck dissection
[72]	case series 16 patients (1 patient with HS)	endoscopic thyroid cancer surgery (new bidirectional approach of video-assisted neck surgery)	bilateral central node dissection (5/16 patients)
[74]	case report (1 patient with HS)	left parapharyngeal ectopic goiter	
[75]	retrospective study 97 patients (2 patients with HS = 2%)		Parapharyngeal lymph node metastases resection
[76]	retrospective study 45 patients (2 patients with H = 5%)	+ (19/45 synchro nous with lateral neck dissection)	selective lateral compartment neck dissection (5/45 bilateral)
[77]	case report (1 patient with HS)	left hemithyroidectomy	
[78]	case report (1 patient with HS)	total thyroidectomy	selective lateral neck dissection
[79]	case report (1 patient with HS)	+	
[80]	case report (2 patients with HS)	minimally invasive video-assisted thyroidectomy (total thyroidectomy)	
[81]	case report (1 patient with HS)	+	
[82]	case report (1 patient with HS)	thyroid microwave ablation	
[83]	case report (2 patients with HS)	video-assisted thyroidectomy	
[84]	case report (1 patient with HS)	thyroidectomy (palliative procedure for anaplastic thyroid carcinoma)	radical neck dissection (right side)
[85]	retrospective study60 patients (1 patient with HS)	thyroid microwave ablation (21/60)	
[150]	case report (1 patient with HS)	+	modified neck dissection type III (right side) + elective dissection at levels II, III, and IV (left side)
[151]	case series 11 patients (3 patients with HS)	+	modified radical neck dissection
**Pediatric population**			
[145]	case report (1 patient with HS)	+	
[146]	retrospective study 16 patients with thyroid cancer (1 patient with HS)	+	cervical lymph node dissection (15/16)
[148]	retrospective study 25 patients with thyroid cancer (2 patients with HS)	+	lymph node dissection (15/25)
[149]	retrospective study 177 thyroid procedures (2 patients with HS)	total thyroidectomies (133/177) hemi-thyroidectomies (40/177)	central or lateral neck dissection (53/177)

Abbreviations: HS = Horner syndrome. The sign “+” means the patient had a thyroidectomy done.

## Data Availability

Data sharing not applicable.

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
