# Peer review of "Horner’s Syndrome and Lymphocele Following Thyroid Surgery"

_jcm, 2023, doi:10.3390/jcm12020474_

Round 1
Reviewer 1 Report
Good work reviewing the complication. The txt could be edited though to list the complication with complications with references in a single paragraph in the introduction section.
I would put in tables to describe the complications that occur with thyroidectomy vs thyroidectomy with central neck dissection
Author Response
Response to Review 1 Comments
Dear Reviewer,
Thank you very much for your time and your effort to review our manuscript.
We are very grateful for providing your valuable feedback on the article.
Here is our response and related amendment that has been made in the manuscript according to your review (marked in yellow color).
Good work reviewing the complication. The txt could be edited though to list the complication with complications with references in a single paragraph in the introduction section.
Thank you very much. This is an interesting approach. However, we respectfully intended to introduce the larger frame of Horner syndrome since the topic might involve different practitioners, not strictly those related to thyroid surgery.
I would put in tables to describe the complications that occur with thyroidectomy vs thyroidectomy with central neck dissection
Thank you very much for your most interesting suggestion which we followed. We really appreciate this idea and we consider that it adds value to our article. We revisited the papers concerning the complication following thyroidectomy and introduced a new Table within the main manuscript.
Table 1. Published papers concerning post-thyroidectomy Horner syndrome (the order of display is according to the main text body; please see references no. 59,60,61,62,63,64,65,70,71,72,74,75,76,77,78,79,80,81,82,83,84,85,145,146,148,149, 150, 151)
Reference number within main text |
Studied population |
Thyroidectomy |
Central neck dissection |
Adult population |
|||
59. |
case report (1 patient with HS) |
+ |
|
60. |
cases series (6 patients with HS) |
+ |
|
61. |
case series 21 patients with different tumors (1 patient with HS) |
#reverse “L” surgical approach |
|
62. |
case report (1 patient with HS) |
+ |
Selective neck region VI dissection |
63. |
retrospective study 1000 patients with thyroid cancer (1 patient with HS) |
robot-assisted endoscopic thyroid surgery using a gasless, transaxillary approach |
Ipsilateral central compartment node dissection (malignant cases) |
64. |
retrospective study 338 patients (1 patient with HS) |
robot-assisted endoscopic thyroid operations using a gasless, transaxillary approach |
Ipsilateral central compartment node dissection (malignant cases) |
65. |
case report (1 patient with HS) |
left hemi-thyroidectomy in addition to ectopic parathyroid gland removal and four-gland parathyroid exploration (no pre-operatory imaging parathyroid adenoma localization) |
|
70. |
case report (1 patient with HS) |
+ (endoscopic thyroid surgery) |
Central lymph node dissection (endoscopic thyroid surgery) |
71. |
case report (1 patient with HS) |
+ left-side minimally invasive video-assisted thyroidectomy |
Neck dissection |
72. |
case series 16 patients (1 patient with HS) |
endoscopic thyroid cancer surgery (new bidirectional approach of video-assisted neck surgery) |
Bilateral central node dissection (5/16 patients)
|
74. |
case report (1 patient with HS) |
left parapharyngeal ectopic goiter |
|
75. |
retrospective study 97 patients (2 patients with HS=2%) |
|
Parapharyngeal lymph node metastases resection |
76. |
retrospective study 45 patients (2 patients with H=5%) |
+ (19/45 synchro nous with lateral neck dissection) |
Selective lateral compartment neck dissection (5/45 bilateral) |
77. |
case report (1 patient with HS) |
left hemithyroidectomy |
|
78. |
case report (1 patient with HS) |
total thyroidectomy |
Selective lateral neck dissection |
79. |
case report (1 patient with HS) |
+ |
|
80. |
case report (2 patients with HS) |
minimally invasive video-assisted thyroidectomy (total thyroidectomy) |
|
81. |
case report (1 patient with HS) |
+ |
|
82. |
case report (1 patient with HS) |
thyroid microwave ablation |
|
83. |
case report (2 patients with HS) |
video-assisted thyroidectomy |
|
84. |
case report (1 patient with HS) |
thyroidectomy (palliative procedure for anaplastic thyroid carcinoma) |
Radical neck dissection (right side) |
85. |
retrospective study 60 patients (1 patient with HS) |
thyroid microwave ablation (21/60) |
|
150. |
case report (1 patient with HS) |
+ |
Modified neck dissection type III (right side) + elective dissection at levels II, III, and IV (left side) |
151. |
case series 11 patients (3 patients with HS) |
+ |
Modified radical neck dissection |
Pediatric population |
|||
145. |
case report (1 patient with HS) |
+ |
|
146. |
retrospective study 16 patients with thyroid cancer (1 patient with HS) |
+ |
Cervical lymph node dissection (15/16) |
148. |
retrospective study 25 patients with thyroid cancer (2 patients with HS) |
+ |
Lymph node dissection (15/25) |
149. |
retrospective study 177 thyroid procedures (2 patients with HS) |
total thyroidectomies (133/177) hemi-thyroidectomies (40/177) |
Central or lateral neck dissection (53/177) |
Abbreviations: HS=Horner syndrome
Thank you very much

Reviewer 2 Report
The authors aim to overview post thyroidectomy Horner syndrome and post thyroidectomy chylous leakage/ lymphocele
on the endoscopic thyroid paragraph - line 172 - please add if TOETVA or Robotic or MIVAT and specify. Did you find data on TOETVA or Robotic?
the authors mention microwave ablation- line 211- should be ablation only -as they describe in the proceeding paragraph RFA, EA, MWA, HIFU etc.
Author Response
Response to Review 2 Comments
Dear Reviewer,
Thank you very much for your time and your effort to review our manuscript.
We are very grateful for your insightful comments and observations, also, for providing your valuable feedback on the article.
Here is a point-by-point response and related amendments that have been made in the manuscript according to your review.
The authors aim to overview post thyroidectomy Horner syndrome and post thyroidectomy chylous leakage/ lymphocele. On the endoscopic thyroid paragraph - line 172 - please add if TOETVA or Robotic or MIVAT and specify. Did you find data on TOETVA or Robotic?
Thank you very much. We mentioned the technique in each study where the information as available.
For instance:
“A study on 3,000 patients who underwent robotic thyroidectomy with axillary approach for thyroid cancer identified the most common complication hypocalcemia (37% of persons had transitory type and 1% of cases were permanent); local collections like seroma (1.73%), hematoma (0.37%), chylous leakage (0.37%), and HS in 0.03% of all cases [137].”
“All types of thyroidectomy are reported to potentially add a risk of developing post-operative HS, from open to endoscopic procedures, including particular resections of lymph nodes and reverse “L” thoracotomy for tumors situated at the level of cervicothoracic junction, including aggressive thyroid carcinoma….[59-64].”
Ref no. 63 – Table 1
63. |
retrospective study 1000 patients with thyroid cancer (1 patient with HS) |
robot-assisted endoscopic thyroid surgery using a gasless, transaxillary approach |
Ipsilateral central compartment node dissection (malignant cases) |
Ref no. 64 – Table 1
64. |
retrospective study 338 patients (1 patient with HS) |
robot-assisted endoscopic thyroid operations using a gasless, transaxillary approach |
Ipsilateral central compartment node dissection (malignant cases) |
“Tang M. et al. published in 2022 the results of 1,213 thyroidectomies pointing out that HS ratio after endoscopic approach is higher when compare to open procedure (0.39% versus 0.29%) [69].”
“Endoscopic thyroid surgery (ETS) – related HS is reported in a few cases reports. For instance, we mention a 31-year-old woman who had a total thyroidectomy done in addition to central lymph node dissection via ETS. Pathological report confirmed a papillary thyroid microcarcinoma with an autoimmune chronic background. She developed HS within the third post-operatory day with progressive remission after a 3-month period of surveillance [70].”
“We mention a few studies reporting chylous/lymphatic leakage, most of them with a small size considering the enrolled population. A series of 12 subjects concerning lateral neck (Vb) dissection via endoscopic surgery with chest-breast approach (CBA) identified one subject (1/12) with the complication [117].”
“A study on 57 patients who underwent left central lymph node dissection via thyroidectomy (CBA) for thyroid cancer found one case (belonging to the subgroup with endoscopic, not open approach) [118]. Another study (N=24) on patients referred for lateral neck dissection at the levels IIA, IIB, III, and IV via endoscopic CBA for malignant thyroid conditions identified 2 cases with chylous leakage (8.3%) [119]. Another retrospective study on 18 subjects with endoscopic thyroidectomy and neck dissection (II, III, IV, and VI) versus 20 patients with open total thyroidectomy showed a similar prevalence of lymphatic drainage (1/18 versus 3/20, p=0.606) [120].”
“Another retrospective study on 18 subjects with endoscopic thyroidectomy and neck dissection (II, III, IV, and VI) versus 20 patients with open total thyroidectomy showed a similar prevalence of lymphatic drainage (1/18 versus 3/20, p=0.606) [120].”
Reference no. 70 – Table 1
70. |
case report (1 patient with HS) |
+ (endoscopic thyroid surgery) |
central lymph node dissection (endoscopic thyroid surgery) |
Reference no. 72 – Table 2
72. |
case series 16 patients (1 patient with HS) |
endoscopic thyroid cancer surgery (new bidirectional approach of video-assisted neck surgery) |
bilateral central node dissection (5/16 patients)
|
The authors mention microwave ablation- line 211- should be ablation only -as they describe in the proceeding paragraph RFA, EA, MWA, HIFU etc.
Thank you very much. We corrected it.

Reviewer 3 Report
-
This narrative review explore data on cervical sympathetic chain injuries and its consequence, Horner's syndrome, as a result of thyroid pathology or surgical intervention. The review is almost clear, comprehensive and of relevance to the field.
I just have a few comments
- In my opinion, the clinical case might be described in the first part of the text, than I would describe the literature.
- I suggest to review the graphical appearance of US images. I suggest to provide graphical image with similar length scale to make changes in lymphocele/chylous leakage
- Paragraph 3 may benefit of a more detailed paragraph partition ( is. Thyroid conditions and HS and lymphocele/chylous leakage related to goiter and iatrogen conditions would reserve two distinct paragraphs for clarity and conciseness.
Author Response
Response to Review 3 Comments
Dear Reviewer,
Thank you very much for your time and your effort to review our manuscript.
We are very grateful for your insightful comments and observations, also, for providing your valuable feedback on the article.
Here is a point-by-point response and related amendments that have been made in the manuscript according to your review (marked in yellow color).
This narrative review explores data on cervical sympathetic chain injuries and its consequence, Horner's syndrome, as a result of thyroid pathology or surgical intervention. The review is almost clear, comprehensive and of relevance to the field.
Thank you very much. We appreciate it.
I just have a few comments. In my opinion, the clinical case might be described in the first part of the text, than I would describe the literature.
Thank you very much. We respectfully choose the overview the literature then introduce the case followed by particular aspects in pediatric population in order to offer a larger frame of these challenging topics which involve multiple medical and surgical aspects of medicine. Thank you.
I suggest to review the graphical appearance of US images. I suggest to provide graphical image with similar length scale to make changes in lymphocele/chylous leakage.
Thank you very much. Indeed, you suggestion is on point. But, unfortunately, as we already mentioned, the child was diagnosed and treated during the dramatic days of strict regulations amid COVID-19 pandemic. That is why, we followed by ultrasound the patient on different devices by different ultrasound specialists depending on the moment we assessed the patient and the feasibility of the procedure amid regulations. Moreover, the latest captures are provided by the family of the patient (her mother) and the patient since ultrasound was performed in her native town which is far from our hospital. We were able to keep in touch with the patient and her family via telemedicine. We already mentioned the turning point in daily practice which is the COVID-19 pandemic.
Thank you very much,
Paragraph 3 may benefit of a more detailed paragraph partition ( is. Thyroid conditions and HS and lymphocele/chylous leakage related to goiter and iatrogen conditions would reserve two distinct paragraphs for clarity and conciseness.
Thank you very much. We performed the paragraph partition.
Thank you
